

# The determinants of chemoreception as evidenced by gradient boosting machines in broad molecular fingerprint spaces

Sammy Sambu

Computational Chemistry, Bartanel Discovery, Zaventem, Belgium

## ABSTRACT

The ability to identify and reject bitter molecules may determine evolutionary fitness. These molecules might be in potentially toxic or contaminated food. Surprisingly, the ability to identify but tolerate or even enjoy bitter foods and medicines may be beneficial. For example, the tolerance of bitterness as a spice or as a medicine may lead to better nutritional, immunological and health outcomes. More recently the ability of intensely bitter compounds to induce innate immune responses to counter infection has inspired the screening of new drugs and the repurposing of safe, known drugs to new uses. These avenues of study may also help to address long-standing questions regarding unexpected side-effects and placebo/nocebo effects. Therefore, to distinguish all these effects ranging from desire to aversion, there is a need to quantitatively determine the concentration thresholds and to position these bitter substances on a unified taste threshold spectrum. Such an understanding may help elucidate the concentration-based molecular drivers for the chemoreceptive response to bitter substances. This article reports the development of a gradient boosting machine (GBM) that enables a direct interrogation of molecular structure with no intermediary chemical properties. Using molecularly engineered simulations, it is shown that potassium acesulfame has a hidden bitterness motif that is centered on the chemoreceptive spectrum uniting bitterness and sweetness molecular motifs. The resultant shifted perception from a touchstone bitterness sensation to a bitter after-taste is attributable to this cached molecular motif.

## INTRODUCTION

Gustation is a critical sense for our daily living. The ability to taste specific compounds is intimately linked to our dietary choices, safety and health.

Human dietary choices have also evolved according to cues drawn from bitter tastes. The adoption of bitter spices into cuisine is also noteworthy since there is a range of spices that bears a distinct bitterness. Some examples include mustard and chervil. It is also notable that traditional vegetables such as the African Spider Plant and the Malabar Spinach are cherished for their bitterness just as much as they are for their robust nutritional value (*Kimiywe et al., 2007*). There is now a substantial collection of research showing

Corresponding author
Sammy Sambu,
Sambu@post.harvard.edu

PeerJ _______________

that the systemic responses to glucose differ between meals containing bitter foods from those without bitter foods. These reports suggest that satiety and the non-diabetic glycemic response can be induced in diabetic respondents when bitter compounds are ingested. These findings appear to support the premise that human feeding response and endocrinology are co-evolved (*Kim, Egan & Jang, 2014*).

"Bitters" have also been used as a flavoring agent for alcoholic beverages globally as means of fortification; in some instances, these bitter cocktails are selected for their medicinal effect e.g., in the historic use of quinine for the preparation of tonics in the tropics. The medicinal and gustatory effects have therefore been conflated in recent times becoming part of the barman's lore (*Barnett, 2012*).

In spite of the unsupported conflation of gustatory and medicinal effects, there is reason to believe that the bitterness receptors (hTAS2R family) may have some links to immunity at a molecular level. It is known that there are extra-oral taste receptors especially in the respiratory system (*Lee & Cohen, 2014*) and in the intestinal mucosa (*Clark, Liggett & Munger, 2012*). Within the respiratory system, ciliated airway epithelia express hTAS2R receptors. These receptors are known to respond to the presence of bacterial quorum sensing molecules by initiating a Ca2+-dependent signaling pathway that increases mucociliary clearance and production of antimicrobial products including peptides and Reactive Nitrogen Species (*Lee & Cohen, 2014*). Gut epithelial tissues are known to contain Tuft cells which have taste-chemosensory capacities enabling the promotion of type-2 immunity in the event of ingress by foreign bodies (*Howitt et al., 2016*).

The ability to distinguish fresh from spoilt food depends on our ability to associate freshness with gustatory cues. This relationship is partly innate and partly learnt (*Zeinstra et al., 2009*). The genetic origins of taste have been confirmed within the human species and more broadly across the animal kingdom. This constitutes the innate bitterness sense. However, the ability to sense a bitter substance is only the beginning as the appropriate learnt response may vary from simple reduction in the amount consumed (e.g., the use of a spice) or the complete avoidance (e.g., aversion). The most intense bitter foods elicit an aversive response which may culminate in an emetic response. Both aversion and emesis are primary immune responses protecting the body from far worse consequences from imbibing poisons (*Palm, Rosenstein & Medzhitov, 2012*).

Not surprisingly, bitter tastes are now demonstrated to have been evolutionarily sectioned according to biogeography. The taste receptor frequency appears to be latitudinally ordered according to the global biogeography. The ability of a species to gustatorily identify poisonous from non-poisonous plants is itself a measure of fitness which is linked to survival (*Chandrashekar et al., 2006*). However, even within a confined latitudinal window, the diversity in the ability to sense the full spectrum of bitterness is now understood to be driven by the zygosity and the epigenetic profile of the taster giving rise to the neologism of a "supertaster" (*Beckett et al., 2014*).

These studies have shown the undervalued importance of bitterness beyond taste but have also raised questions on the precise molecular theory driving bitterness thresholds and responses. Similarly, testing and design principles for bitter compounds require refinement against the often-noisy clinical data. In this regard, the focus of this article is the

construction of model forms that may permit the identification of the relevant molecular space and the structural heuristics for further investigation.

## MATERIALS & METHODS

### Metadata collection

Data was sourced from previous studies by *Meyerhof et al. (2010)* of molecular receptive thresholds drawn from the calcium-signaling responses of hTAS2R transfected cells. Cells were designed to express the hTAS2R epitopes on the cell surface, exposed to the bitter compounds, calcium-sensitive signaling dye and an inhibitor for anion transport (cellular vitality tests were co-evaluated alongside the hTAS2R-mediated calcium signaling). The hTAS2R receptors were coupled to intracellular calcium signaling by the chimeric G-protein subunit, G$\alpha$16gust44 (*Ueda et al., 2003*). Adequate controls were provided by means of empty transfection vectors (*Meyerhof et al., 2010*).

#### *Relative Quinine Index (RQI)*

The RQI is calculated by dividing the detection concentration of a given compound and dividing it by the detection concentration of quinine. The RQI helps to organically communicate how bitter a compound is because quinine is a familiar compound commonly used to make tonic water (a consumer product). Therefore, a molecule that is more bitter than quinine will have a an RQI less than one (the converse is true).

#### *Sample Size & Data Description*

There were in total 82 bitter compounds. These compounds were pre-determined to be bitter based on psychophysical tests (*Meyerhof et al., 2010*). The RQI ranges from 0.00013 to 99.9 representing almost six orders of change in magnitude.

### Generation of molecular structures

Structures were generated using ChemSpider.

### Variable transformation

Molecular structures were checked for consistency before they were used to generate descriptors using R ChemoInformatic packages: Rcpi (*Cao et al., 2015*), ChemmineR (*Cao et al., 2008*) and ChemmineOB (*Horan & Girke, 2013*). Components include: molecular fingerprints (graph, FP4, MACCS)', electron structure, spatial and topological descriptors. See Table 1 for the descriptor set.

### Chemical scaffold determination

The chemical scaffolds were identified using Scaffold Hunter (*Wetzel et al., 2009*). Briefly, chemical identifiers, Names, SMILES and RQI were imported into an HSQLDB database. Thereafter, scaffold clouds and tree maps were generated to represent the inner structural relationships between members of the chemical library. Scaffolds were obtained by using a deterministic structural reduction process that prunes terminal branches revealing inner shared structures referred to as scaffolds. Scaffolds were clustered via the sequential agglomerative hierarchical non-overlapping clustering (SAHN) algorithm (*Anderberg, 2014*; *Schäfer et al., 2017*).

**Table 1** **A representative summary of features that contribute significant information to the model.** Values are obtained after removal of low or zero-variance columns from the original input space.

| Descriptor class | Function | Columnar contribution | Rationale |
|---|---|---|---|
| Electro-topological | extractDrugEstateComplete | 8% | Charge distribution affects binding efficiency and stability |
| Structural | extractDrugMACCSCompleteextractDrugOBFP4 | 60% | |
| Geometric | extractDrugGraph | 32% | Molecular shape may affect ligand docking |

## Description of the SAHN algorithm

The algorithm has four main steps: first, the chemical structures are fingerprinted using bit arrays that capture both existing and missing chemical features. The chemical fingerprints extended with descriptors are then used to calculate a similarity matrix. Pairs of similar molecules are used to create nodes in a sequential fashion that ultimately captures all the members of the dataset in a single mathematical construct (this is the linkage routine). Finally, visual representations of the data are presented using dimensional reduction methods allowing users to have compact slices of the data conveying the structural chemical motifs in a semantically compelling snapshot (*Anderberg, 2014*; *Schäfer et al., 2017*).

## Model development

The computational technique had three steps. The first step was to transform the molecular SMILES into structures and generate the descriptors for structure, electronic state and topology. The second was to generate a high-variance reduced form of the molecular structures capturing the input space. This step involved removing the low and zero-variance vectors in the input space. Low and zero variance vectors are removed because they are uninformative and therefore do not contribute to the determinacy of the system. The third step involved using a gradient boosting machine (GBM) to regress the bitterness thresholds against the input space. A GBM was chosen because it generates initially weak learners and subsequent stronger learners will only improve on weak learners in areas of chemical space where residuals are large. When applied to this bit array representation of molecular descriptors, they will be run on an eight-fold ($8\times$) cross-validated train-test sample split. The training sets are validated by comparing the resulting trained model predictions against the experimental data in the test set. The chosen model will have the lowest overall cross-validated error. The hyperparameters used were: 199 trees, maximum tree depth of 7 levels; minimum tree depth of 2 rows and a learning rate of 0.2 (detailed information included in the code attachments). The choices are driven by the rate of convergence of the algorithm; in turn convergence is driven by the attainment of a sustained minimum in the residuals of the model (see Fig. 1). Additionally, the learning rate served to minimize the test set residuals which means that the overall model performance is within specified metrics. While this means that the convergence is slowed, the other chosen parameters balance this decreased velocity with an aggressively parameterized chain of increasingly complex learners. Model selection was based on a set of 4 values: Coefficient of Determination (R.SQ.), model error, convergence and sustained metrics during validation. The full

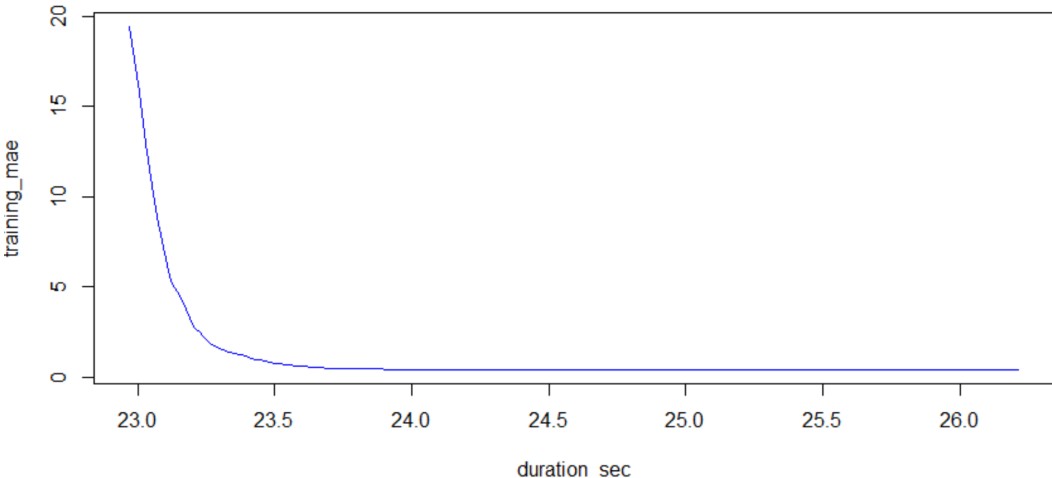

**Training Scoring History**

**Figure 1** **Training Mean Absolute Error plotted against the Training Process Time in seconds.** The graph shows a rapid convergence to a low error. X-validated model metrics: $R^2$ (99.3%); RMSE (2.34); Mean Response (15.39).

dataset and the outlined code have been provided and can be run on R version 3.5.0 or higher.

## RESULTS

### Model generation

The model convergence was demonstrably quick having constructed the input space using molecular fingerprints and descriptors as shown in Fig. 1. Model convergence is directly linked to the choice of hyperparameters which accelerate the reduction in residuals to a sustained minimum. The residuals' magnitudes are captured on the $y$-axis while the algorithm's process time is captured on the $x$-axis in seconds. The attainment of the minimum is shown in the asymptote of the residual-duration curve.

Molecular patterns have previously been observed amongst bitter molecules when contrasted against an intuitive diametrical opposite (sweet molecules). This observation was proven true when molecular changes to moieties on the sweet molecule gave bitter molecules (*Belitz et al., 1983*). Therefore, the molecular structure was used to provide structural information divisible into three areas: geometric/connective, electro-topological and structural representations. A summary of the descriptor choice is provided in Table 1.

The RQI was then regressed against the input space generating a cross-validated model with the following metrics: Coefficient of Determination ($R^2$) (97%), Root Mean Square Error (RMSE) (2.34) for a Mean Response (15.39). The model accuracy is shown in Fig. 2 by plotting the predicted vs. the empirical bitterness index. The points are straddled across the identity line ($y=x$) showing the model tries to match the empirical reality and the errors (deviations away from the identity line) are random. Overall, the mathematical construct confirmed the existence of a robust deterministic relationship between the input
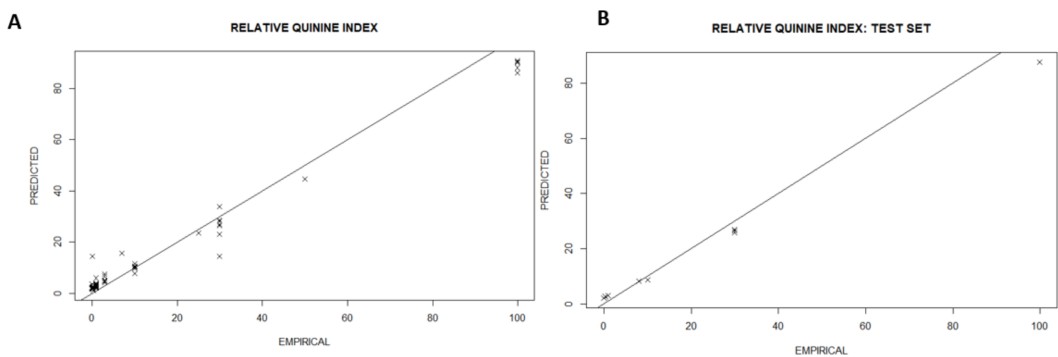

**Figure 2** **All-in model showing the agreement between predicted bitterness index and the empirical value with all values in RQI for both the training (A) and the test set (B).** Model fidelity is maintained in both the training (A) and the test set (B). X-validated model metrics: $R^2$ (99.3%); RMSE (2.34); Mean Response (15.39).

space and the RQI. This is consistent with previous observations of the strong relationship between molecular structure, molecular formula and geometry (*Belitz et al., 1983*) because both the model and observational treatments map from the same foundational chemical features onto measures of bitterness with observations being descriptive while the model is a compact mathematical representation of the same insights.

## DISCUSSION. CAN WE DEVELOP HEURISTICS FROM THE STRUCTURAL INFORMATION?

The observations made in Table 1 are striking given that they offer an abstraction of the drivers of bitterness with a moiety-based adjustment to contrast between configurations on specific molecular structure with shared motifs. Given that the model could sufficiently predict these changes, it can be inferred that the full dataset should lead us to similar conclusions. From Fig. 3, the amine-containing groups contribute greatly to the scaffold cloud to the tune of 8X relative to the sulfinyl groups. This is consistent with previous observations for alkylamines, amides and azacycloalkanes (*Belitz et al., 1983*). The representational model predicts that molecules with pyridyl and amine substructures will have significant bitterness quotients. The centrality of amine-containing bitterants appears to demonstrate a path of modification towards molecular clusters with increasing aversive gustatory qualities as shown in Fig. 4. Clustering approaches are especially vivid where hierarchical and progressive modification are concerned. Not surprisingly, these modifications allude to the origins of bitter substances being the arms-race between plant and animal kingdoms and the search for optimality between fitness targets of protection and dispersal. These observations raise the hope that further research into the bitterness-driven screening libraries and the extensional indications for known and well-tested drugs can be expected to be fecund.

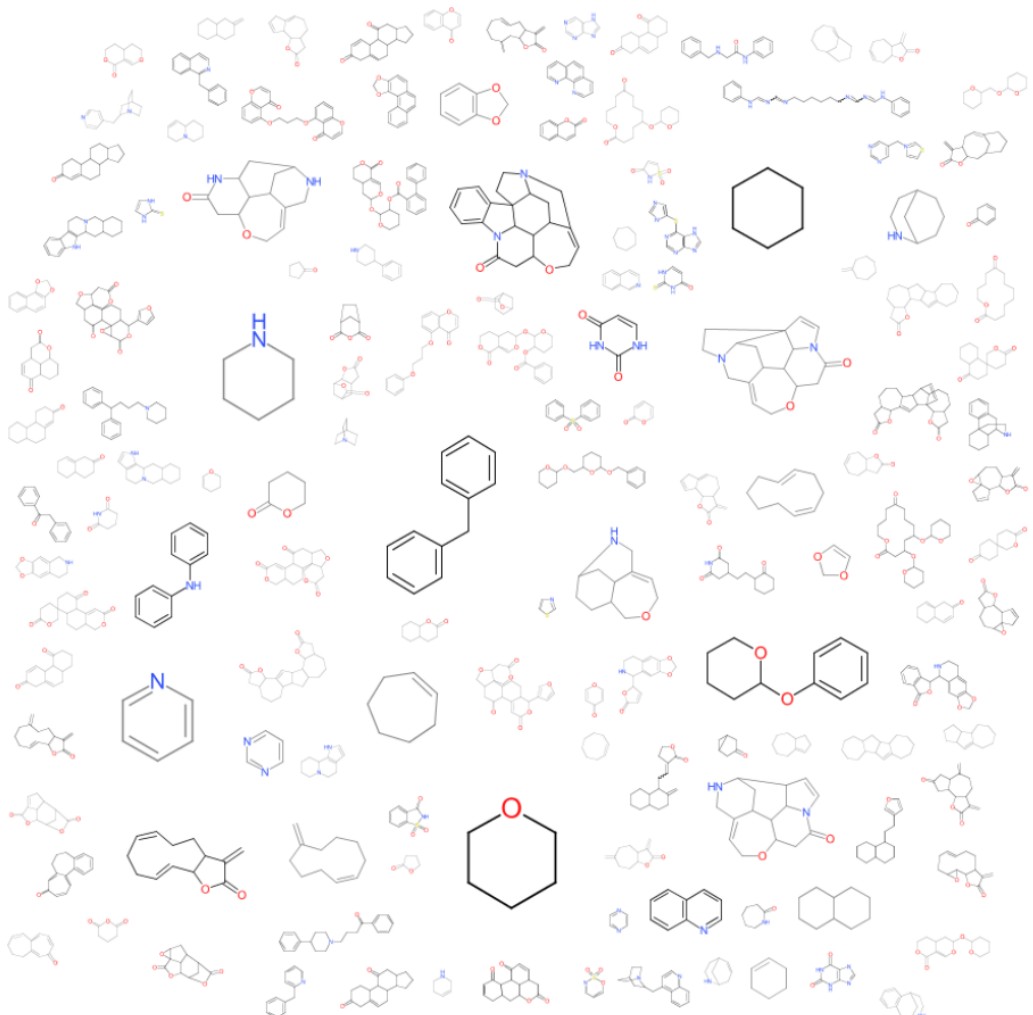

**Figure 3  A visualized structural contribution to the data set.** A cloud visualization indicates that the parametric coefficients that drive bitterness are greater for amine groups than they are for the sulfinyl groups with a greater than 8×preponderance.

## WHY MIGHT POTASSIUM ACESULFAME (ACEK) HAVE A BITTER AFTER-TASTE?

The model was used to explore what changes drive the bitterness threshold of known compounds. AceK has a bitter aftertaste but the existence of some bitterness signaling (high panelist variance) throughout the taste experience indicates that the molecule possesses inherent bitterness (*Kuhn et al., 2004*). Using the model, the hypothesis could be tested by changing moieties that are likely to be transformation sites. The new molecular structures are then used as inputs into the GBM whose output is the *predicted RQI* which represents a bitterness threshold relative to quinine (the chosen bitterness standard). On Table 2, AceK has a respectable bitterness threshold. The RQI does not change meaningfully across similar molecules with a -CNSO- motif; they all have high RQI (i.e.,
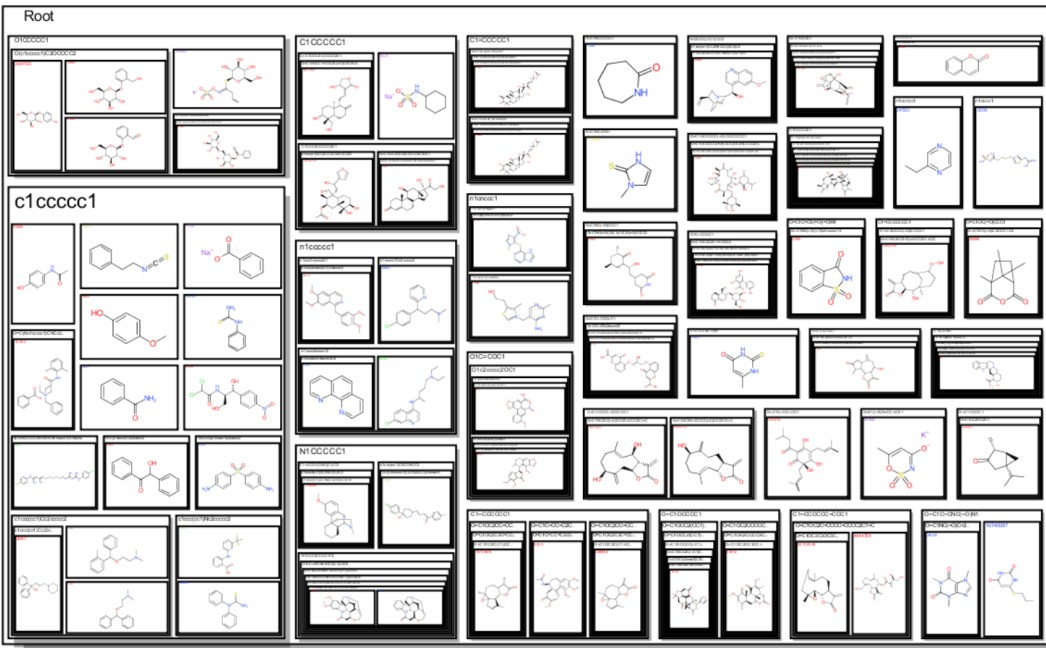

**Figure 4  A structure-based tree map showing an increasing tendency for amine containing centers to the right and to the bottom.** These representations emphasize the significance of the -NCS- motif much like the GBM model and the scaffold cloud.

they are less bitter than quinine). In contrast, maintaining an -NCS- (thiocyanate) motif alone lowers the bitterness threshold even to below that of quinine as exemplified by the PTC (phenylthiocarbamide) standard (*Meyerhof et al., 2010*). This finding may indicate that molecular structure is critical to bitterness sensing and that motifs can be altered by introduction of additional polar atoms to the identified molecular 'trigger motif' to increase the RQI. Additionally, from a ''molecular structure'' perspective there are gradations of taste uniting bitter and sweet on the same scale. Where such gradations meet, as in the AceK case, the hidden bitterness motif is likely altered resulting in a shifting of perceptions to after-tastes rather than a touchstone bitterness sensation. This model-driven conclusion is supported by the observation that the heteromeric sweetness receptor hTAS1R2–hTAS1R3 and bitterness receptors hTAS2R43 and hTAS2R44 are all activated by AceK at higher concentrations for the sweetness receptors implying bitterness would be recognizable at lower concentrations typical of a post-evacuation state within the inundated gustatory bulb i.e., as an after-taste (*Kuhn et al., 2004*). These *in vitro* observations confirm the ability for the AceK 'master' molecular key to unlock both sweet and bitter sensations in that order (*Kuhn et al., 2004*).

## EXAMINING MYCOTIC COMPOUNDS: EVIDENCE OF EVOLUTIONARY COOPERATIVITY

It is observable that homologous compounds demonstrate an evolutionary pressure exerted by saprophytes and autotrophs against herbivorous and omnivorous heterotrophs. A fine

**Table 2 A table showing the estimated and empirical RQI of AceK and molecular similars.** In column 1, the molecule possesses an -NCS- motif and a low RQI. In contrast the molecules in columns 2 & 3 possess a -CNSO- motif and high RQI values. Empirical values are provided as detection ranges from the mean and standard errors of the hTAS2R family receptors.

**ESTIMATED BITTERNESS**

| Name | phenylthiourea | AceK | 1,2-Benzisothiazol-3(2H)-one, 1,1-dioxide |
|---|---|---|---|
| Smiles | C1CCC(CC1)NC(=S)N | CC1=CC(=NS(=O)(=O)O1)[O-].[K+] | C1CCC2C(C1)C(=O)NS2(=O)=O |
| |  |  |  |
| Reason/Change | -NCS- Motif | -CNSO- Motif | -CNSO- Motif |
| Predicted RQI | 0.01 | 26.00 | 8.00 |
| Empirical RQI | 0.001–0.01 | 20–42.5 | 2.5–7.5 |

example is the link between phenylketonuria and ochratoxin A; the former being linked to fetal protection against the latter (*Woolf, 1986*). Therefore, bitter ochratoxin A is linked to changes in the human genome driven by its toxicity (*Woolf, 1986*). Additionally, for the same cognate molecular set, the detection threshold (RQI) for the poisonous strychnine variant molecule (LD50 at 5 mg; RQI at 0.01) is lower than that of its edible and sometimes therapeutic counterpart brucine (LD50 at 1,000 mg; RQI at 1) (*Meyerhof et al., 2010*). These representative LD50-RQI relationships are nonlinear. Much interest in this observation has been driven by the desire to find more powerful and durable antifungals, antibiotics and antivirals. Therefore, the case study here looks to compare cognate molecular sets to identify which among them are likely to fall in the edible-therapeutic group. More specifically, ergolines are of interest given that their somatic and psychiatric effects can span the spectrum of beneficence to toxicity.

We examined the class of ergoline mycotic chemistries known to have pharmaceutical value in humans. These chemistries are known to have vascular (*Martin & Dumoulin, 1953*) and nervous system effects (*Coward et al., 1990*). They are valued for treating a range of conditions including post-partum bleeding (*Martin & Dumoulin, 1953*) and migraines (*Lance, Anthony & Somerville, 1970*). The molecular structures of three ergolines were entered as inputs, converted to chemical fingerprints and descriptors and subsequently predicted by the GBM to have quantified predicted RQIs (column 3, Table 3). Looking at Table 3, the demarcation between the compounds on the toxicity measures (LD50) broadly matched with the RQI as predicted by the model. Low RQI thresholds correspond with low LD50 thresholds supporting the going hypothesis that evolutionary directions for human bitterness receptors appear to follow the surrounding environmental pressures. This motivates the derivation of a ratio being the LD50-to-RQI (toxicity-bitterness) ratio. This Toxicity-Bitterness Ratio approximates 1,500 (ergometrine), 150 (methylergometrine) and 15 (methysergide). This shows an intervallic spread between each evolutionary terminus uniting human and mycotic adaptations. There is a non-linear decrease in the toxicity-bitterness ratio which is attributable to the non-linearity of psychophysical (*RQI-Concentration*) (*Meyerhof et al., 2010*) and toxicity (*LD50-Concentration*) (*O'Brien,*

**Table 3  Tabulation of the Ergoline class showing the sharp demarcation between the compounds on the toxicity measures (LD50) matched with the RQI as predicted by the structural model.** Low RQI thresholds correspond with low LD50 thresholds confirming the going hypothesis that evolutionary directions for human bitterness receptors follow the surrounding environmental pressures.

| Structure | Name | Predicted RQI | LD50.human (ug/kg) |
|---|---|---|---|
|  | Ergometrine | 10 | 15000 |
|  | Methylergometrine | 4.45 | 667 |
|  | Methysergide | 1.85 | 28 |

*Chooprateep & Homkham, 2009*) curves for compounds including ergolines which are phenomenologically modeled using non-linear regressive methods.

## CONCLUSIONS

In conclusion, a molecular-theoretic approach to predicting bitterness thresholds for the human T2R receptor has been developed demonstrating exquisite model quality diagnostics. The outcomes may be usable in the testing and design of bitter compounds targeted at taste-chemosensory receptors. Model assessments have also allowed us to identify the importance of electro-topological, structural and geometric properties of the molecular space. Further, the model was usable in developing verifiable structural heuristics for bitterness, explaining aftertaste sensations chemometrically and separating toxic from non-toxic therapeutic molecular cognates. It is proposed that future work may focus on the mechanistic drivers of receptor-driven immune responses addressed to the greater challenge of identification of scaffolds for immunotherapeutic small molecules and next-generation adjuvants.

### Funding
The author received no funding for this work.

### Competing Interests
The author declare that there are no competing interests.

### Author Contributions
- Sammy Sambu conceived and designed the experiments, performed the experiments, analyzed the data, contributed reagents/materials/analysis tools, prepared figures and/or

tables, performed the computation work, authored or reviewed drafts of the paper, approved the final draft.

## Data Availability

Data is available in the Supplemental Materials.

## Supplemental Information

Supplemental information for this article can be found online at http://dx.doi.org/10.7717/peerj-ochem.2#supplemental-information.

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
