# Peer review of "The determinants of chemoreception as evidenced by gradient boosting machines in broad molecular fingerprint spaces"

_PeerJ Organic Chemistry, doi:10.7717/peerj-ochem.2_

## Round 0.1 · original submission · Major Revisions

Thank you for your submission. Please revise the manuscript according to the reviewers feedback.

Reviewer 1 ·

Basic reporting

A very good well-written paper that will attract the attention of scientists. The basic premise is well-chosen and the methodology and execution justifies the hypothesis

Experimental design

The design is sound and I believe that the author has justified his hypothesis well

Validity of the findings

I believe the article to be well-researched; the investigations are sound> These findings could find broad utility in research

Additional comments

I like the basic premise of the paper. I found it to be interesting .

·

Basic reporting

Numbered points are in order of precedence, bullets signify points to be
addressed in no order of precedence.

1. The abstract does not contain details of the study and needs to be re-written
to include the results. Specifically, the AceK insights should be a part of
the abstract.
2. Machine learning in the title is too vague. Consider being more specific,
e.g. "as evidenced by machine learning" -> "as evidenced by gradient boosting models in molecular fingerprint spaces"

- 156: "Well known" should be backed by a citation.

Minor Spelling and Grammar issues (indicative, non-exhaustive):

- 15-18: the sentence is overly long
- 30: "the cues drawn" -> "cues drawn"
- 36: "It appears" -> "It has been reported"
- 41: "used as a flavoring" -> "used as a flavoring agent"
- 47: Unclear why nevertheless is used
- 54: "have a taste-chemosensory capacity" -> "have taste-chemosensory capabilities"
- 90: "a ChemSpider" -> "ChemSpider"

Experimental design

Numbered points are in order of precedence, bullets signify points to be
addressed in no order of precedence.

1. 98-100: The scaffold process needs to be described in more detail, including
the SAHN algorithm and how this is relevant to reduce and cluster the
underlying chemical structures.
2. 109: Why has the gradient boosting model been used? How has the model been
generated? What is the rationale behind the parameters? How was the data
validated? What were the hyper-parameters? Figures 1. and 2. need more of an
explanation in the text.
3. What is the relation between the RQI and LD50.human? (Table 3) Has the LD50
been correlated with the RQI in this study or others? Has it been factored
into the model development? The scaling is clearly non-linear (RQI v/s LD50)
so the correlation should be described in more detail.
4. 82-89: The data-set used should be described in more detail, including at
least the description of how many data points were obtained.
5. 104-111: This section should be expanded on with an example, detailing the
transformation, descriptor generation and other transformation metrics.
6. 107-108: Why have the low and zero variance vectors been removed?
7. 127: Further comment on why the regressed model correlates well with the
observations would add to the manuscript.

Validity of the findings

Numbered points are in order of precedence, bullets signify points to be
addressed in no order of precedence.

1. Figure 2 is unclear. What do the hollow dots and the filled in dots signify?
2. 184: 3 data points which do not show linear ordering is insufficient to
confirm a hypothesis. Consider using different language if more data is not available.
3. 130: It is unclear as to how the Predicted RQI performs here without
information on the actual RQI. Table 2 should include the actual RQI.

- 162: Though a reasonable conclusion from the data, any supporting evidence
would help in terms of establishing the distinction between after tastes and
initial taste sensations.

Additional comments

The authors have explored computationally, an abstracted input space of chemical
structures and have regressed over a model (details unclear) to obtain insights
into the RQI index, a measure of bitterness. The work presented is very
interesting and the details surrounding the necessity of the study have been
presented in a compelling manner. The conclusions drawn from the data are also
largely consistent, not-with-standing the issues listed above. The model
description is inadequate, but after these have been addressed this will be a
very timely addition to the existing literature.

·

Basic reporting

The reporting style and flow of the language in the manuscript make it an engaging and compelling read. However, despite being well-written in general, there are a few minor grammatical issues/typos to be addressed, listed below in no particular order.

1. The first line of the abstract (lines 15:18) is a little long and could be split into two sentences. Similarly, the sentence in lines 98:103 could be broken into two.

2. The following replacements are suggested: line 17 ("have" --> "has"); line 19 ("ability for intensely bitter molecules" --> "ability of intensely bitter molecules"); line 31 ("are a range" --> "is a range"); line 35 ("showing the systemic responses..." --> "showing that the systemic responses..."); line 47 ("to believe the bitterness receptors..." --> "to believe that the bitterness receptors..."); line 69 ("The ability for" --> "The ability of"); line 75 ("questions on the precise a molecular theory" --> "questions regarding the precision of a molecular theory"); line 90 ("using a ChemSpider" --> "using ChemSpider"); line 131 ("given they" --> "given that they"); line 133 ("Give the model could sufficiently..." --> "Given that the model could sufficiently..." or "Since the model could sufficiently..."); line 134 ("it reasons" --> "it stands to reason" or "it is reasonable to suppose" or "it can be inferred"); line 136 ("relative the" --> "relative to the");

3. The expansions of the acronyms RQI (line 124), R2 (line 125), RMSE (line 125) should be included once.

Experimental design

I was wondering how the bitterness index quantifies the gradations of bitterness, and more details of the same could be provided in the Methods or Results sections.

Validity of the findings

The sample size of the data, controls and scientific methods adopted in this work are adequate. The bitterness index and the relationship between RQI and the results could be expounded further.

Additional comments

I think the manuscript has been written in professional, unambiguous language, with several compelling points made. Slightly more description in the Methods and Results might improve the manuscript further.

---

## Round 0.2 · accepted · Accept

Thank you for your contribution which is now Acceptable